# Heat stress induces phage tolerance in *Enterobacteriaceae*

Fan Zhang[1], Hao-Ze Chen[1], Bo Zheng[1], Liang Huang[1], Ye Xiang[1,2,3,4], Jing-Ren Zhang[1,2], Jia-Feng Liu[1,2]*

[1]Center for Infection Biology, School of Basic Medical Sciences, Tsinghua University, Beijing, China; [2]Tsinghua-Peking Joint Center for Life Sciences, Beijing, China; [3]SXMU-Tsinghua Collaborative Innovation Center for Frontier Medicine, Taiyuan, China; [4]Beijing Frontier Research Center for Biological Structure, Tsinghua University, Beijing, China

## eLife Assessment

This **important** study analyzes the effect of heat treatment on phage-bacterial interactions and **convincingly** shows that prior heat exposure alters the bacterial cell envelope, enhancing persistence and bacterial survival when exposed to lytic phages. The study will interest researchers working on antibiotic resistance, tolerance, and phage therapy.

*For correspondence:
jfliu@mail.tsinghua.edu.cn

Competing interest: The authors declare that no competing interests exist.

**Abstract** Antibiotic resistance and tolerance present significant challenges in global healthcare, necessitating alternative strategies such as phage therapy. However, the rapid emergence of phage-resistant mutants poses a potential risk. Here, using *Klebsiella pneumoniae* ATCC 43816 and its lytic phage Kp11 as a model system, we investigated bacterial persistence against phages, characterized by heterogeneous survival, analogous to antibiotic persistence. We found that heat treatment enhanced persistence and increased bacterial survival under phage exposure, subsequently promoting the evolution of phage resistance. Further experiments demonstrated that heat stress leads to a reduction in envelope components, thereby inhibiting phage DNA injection. Additionally, this heat-induced reduction resulted in systematic alterations in envelope stress responses, rendering bacteria tolerant to the antibiotic polymyxin while making them hypersensitive to pH changes and immune clearance. Our findings provide novel insights into bacteria-phage interactions and highlight potential challenges in implementing phage therapy in clinical settings.

## Introduction

Antibiotic resistance poses a significant threat to global public health, emphasizing the urgent need for innovative strategies to combat bacterial infections (*Luong et al., 2020*). Phage therapy has emerged as a promising alternative, yet the rapid emergence of phage-resistant bacterial mutants challenges its effectiveness (*Oechslin, 2018*; *Salmond and Fineran, 2015*; *Strathdee et al., 2023*).

Beyond resistance, antibiotic tolerance presents a critical concern, enabling bacteria to survive high concentrations of antibiotics (*Balaban et al., 2019*; *Balaban et al., 2004*; *Brauner et al., 2016*), which can further promote the evolution of resistance (*Levin-Reisman et al., 2019*; *Levin-Reisman et al., 2017*; *Liu et al., 2020*). This raises an important question: Could analogous persistence and tolerance arise in response to phage infections, thereby contributing to the development of phage resistance?

**Figure 1.** Bacterial persistence against phage infection. (**a**) Schematic illustration of the experiments demonstrating the existence of phage persisters. (**b**) Viable colonies of *K. pneumoniae* ATCC 43816 at indicated time points post-infection with phage Kp11 (multiplicity of infection, MOI ~100), grown on both phage-containing (+phage) and phage-free (-phage) plates. Curves represent the mean ± standard deviation (s.d.) of three biological replicates. (**c**) Survival fraction after 40 min of phage Kp11 treatment for the ancestral strain ATCC 43816 and descendants of persisters. Data are presented as mean ± s.d. of three biological replicates; ns indicates not significant (p=0.6131 and 0.6415). (**d**) Viable colonies of *KLY* and its high-persistence derivative strain *metG\** that survived treatment with the antibiotic ertapenem (5 µg/mL, 20×MIC) and phage T4 infection (MOI ~100). Curves show the mean ± s.d. of three biological replicates.

The online version of this article includes the following source data and figure supplement(s) for figure 1:

**Source data 1.** Data used for graphs presented in *Figure 1b–d*.

**Figure supplement 1.** Complete genome of lytic phage Kp11 against *Klebsiella Pneumoniae* (GenBank: ON148528.1).

**Figure supplement 2.** Susceptibility to phage T4 is similar for *KLY* and longer lag-time strain metG\*.

**Figure supplement 2—source data 1.** Data used for graphs presented in *Figure 1—figure supplement 2a and b*.

## Results

### Bacterial persistence against phage infection

To address this question, we investigated the killing dynamics of *Klebsiella pneumoniae* under treatment with the lytic phage Kp11 (GenBank: ON148528.1, *Figure 1—figure supplement 1*; *Huang et al., 2024*). In analogy with the concept of antibiotic persistence, the majority of bacteria were rapidly eliminated following lytic phage treatment, while a small fraction (~$10^{-5}$ to $10^{-6}$) survived, resulting in a bi-phasic killing curve (*Figure 1b*). However, the bacteria that survived likely comprised

both persisters and resistant mutants. To quantitatively distinguish between resisters and persisters, colonies of surviving bacteria post-phage treatment were individually plated on media with and without phage. The fraction of persisters was estimated by subtracting the number of resisters from the total colony-forming units (CFU; *Figure 1a and b*). Subsequent killing experiments on the descendants of persisters revealed a comparable survival fraction to the ancestral population, suggesting that this survival was not attributable to genetic mutation but rather to phenotypic heterogeneity, that is phage persistence (*Figure 1c*).

Canonical antibiotic persistence is characterized by an extended lag time or slow growth rate (*Balaban et al., 2004*; *Brauner et al., 2016*). However, previous studies showed that the high antibiotic persistence *Escherichia coli* strain *hipA7* did not exhibit enhanced survival when exposed to lytic phages compared to its sensitive ancestor (*Pearl et al., 2008*). As there currently lacked a well-defined high-persistence variant of *K. pneumoniae*, we turned to the *E. coli* strain *KLY* and its high-persistence derivative *metG** and challenged these two well-studied strains with T4 phage, which was known for its lytic activity against *E. coli*. The observed similar survival rates suggested that increased lag time does not confer protection against phage infection (*Figure 1d*, *Figure 1—figure supplement 2*).

## Phage tolerance induced by heat stress

Given that antibiotic persistence can be triggered by stresses such as starvation (*Betts et al., 2002*; *Nguyen et al., 2011*), we explored whether phage persistence might similarly be affected or not. *K. pneumoniae* was subjected to various stress conditions prior to phage treatment. Remarkably, pre-treatment at elevated temperatures significantly reduced *K. pneumonia*'s susceptibility to lytic phage infection (*Figure 2a*). Killing assays revealed that heat-treated bacteria exhibited a reduced killing rate following phage exposure (*Figure 2b*), with effects being both time- and temperature-dependent (*Figure 2—figure supplement 1*). Moreover, infection curves indicated that heat-treated bacteria relapsed more frequently, suggesting a higher probability of resistance mutant emergence (*Figure 2c*).

To determine whether heat treatment directly conferred increased phage resistance, we quantified the fraction of phage-resistant mutants before and after heat treatment. Survivors from the phage killing assay were collected and plated on both phage-containing and phage-free plates, with only resistant colonies detectable on the phage-containing plates. Our analysis revealed a negligible presence of resistant mutants in both groups (*Figure 2d*). Consistent with this, mutation rates estimated using Delbrück-Luria tests (*Luria and Delbrück, 1943*) confirmed no significant difference (*Figure 2e*). Furthermore, although the survival rate of heat-treated bacteria substantially increased after 2 hours of phage infection, the proportion of phage-resistant mutants remained comparatively small (*Figure 2d*). These findings indicated that the enhanced defense against phage infection observed following heat treatment was not due to an increase in phage-resistant mutants, but rather reflected a phenotypic boost in survival analogous to antibiotic persistence (*Brauner et al., 2016*). Therefore, we propose that the observed increase in resistance frequency is due to the cumulative effect of a net increase in survival, similar to how antibiotic tolerance promotes resistance.

Inspired by this concept, we conducted experiments using a very small bacterial inoculum (~$10^3$) to minimize the presence of preexisting resistant mutants, ensuring that any observed resistance arose during phage infection. To maximize phage-bacteria interactions, we used a high multiplicity of infection (MOI ≥1) and a small culture volume (100 μL). After 24 hr of incubation with or without heat treatment and phages, untreated bacteria were completely eradicated, resulting in clear wells (*Figure 2f*). In contrast, turbidity – defined as an $OD_{600}>0.9$ – was observed in wells containing heat-treated bacteria co-cultured with lytic phages, indicating the emergence and growth of resistant mutants during treatment (*Figure 2f*).

Microscopic observations further revealed that heat-treated bacteria could divide under phage pressure, while untreated bacteria were lysed by phages before dividing (*Figure 2g* and *Figure 2—videos 1 and 2*). This indicates that the tolerance induced by heat treatment facilitates the emergence and evolution of phage resistance, as heat-treated bacteria underwent more division events under phage pressure, ultimately increasing the likelihood of developing phage resistance (*Levin-Reisman et al., 2017*; *Liu et al., 2020*).

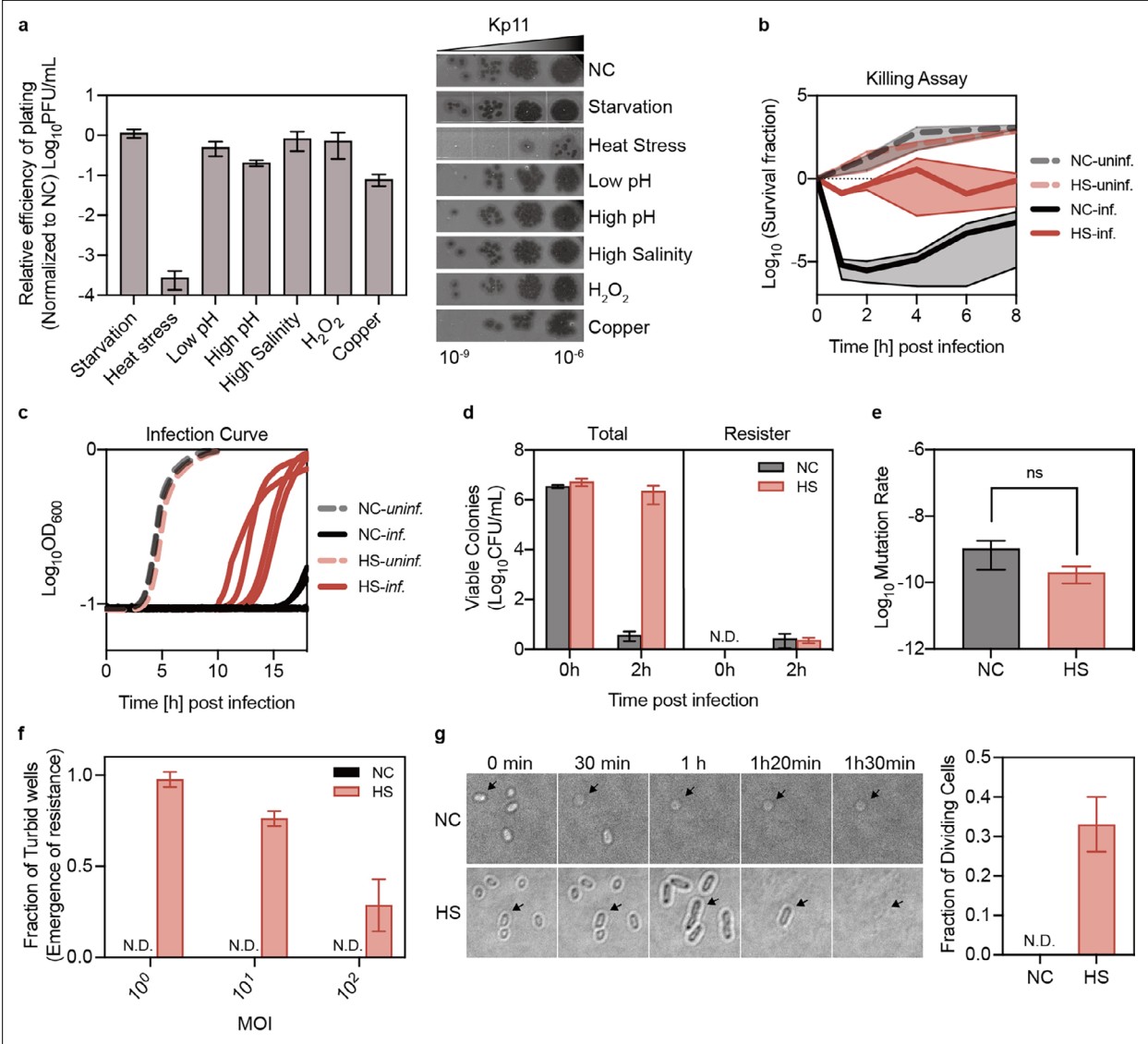

**Figure 2.** Phage tolerance induced by heat stress in *Klebsiella pneumoniae*. (**a**) Plaque-forming units of the infection drop assay in *K. pneumoniae* ATCC 43816 following various pre-treatments. The morphotypes of bacteria corresponding to each stress pre-treatment are indicated: starvation for 36 hr in 0.9% NaCl; heat stress for 1.5 hr at 50 °C; and treatment for 1 hr with low pH (pH 3.2), high pH (pH 10), high salinity (5% NaCl), oxidative damage (10 mM $H_2O_2$), and copper (5 mM $CuSO_4$). Data are presented as mean ±s.d. of three biological replicates. NC: Negative Control, bacteria without high-temperature treatment prior to phage infection. (**b**) Survival of *K. pneumoniae* ATCC 43816 pre-treated with heat and subsequently infected with phage Kp11 (MOI ~100). Curves represent the mean ±s.d. of five biological replicates, with the solid line indicating the mean and the shaded area representing the s.d. of the mean. HS: Heat Stress, bacteria subjected to high-temperature treatment at 50 °C for 90 min prior to phage infection. (**c**) Growth dynamics of liquid cultures of *K. pneumoniae* ATCC 43816, with and without heat treatment, infected with phage Kp11 (MOI ~100) at 37 °C. Curves show the mean of four independent replicates. (**d**) Bacteria from the killing assay (panel b, time 0 and 2 hr) were collected and plated on plates with and without phage Kp11 ($10^8$ PFU). No phage-resistant colonies were detected at time 0 (N.D.: Non-detectable), and similar numbers of phage-resistant colonies were observed at time 2 hr in both populations. Data are presented as mean ±s.d. of three independent replicates. (**e**) Mutation rates of *K. pneumoniae* ATCC 43816 with and without heat treatment are shown. Data are presented as mean ±s.d. from five independent replicates and 14 plates for each replicate; ns indicates non-significant (p=0.3194). (**f**) Bacteria (~$10^3$ CFU) with and without heat treatment were cultured with phage Kp11 at various MOIs of 1, 10, and 100 for 24 hr, with the fraction of turbid wells shown indicating resistance emergence. Data are presented as mean ±s.d. of three independent replicates and 14 wells for each replicate, and no turbid wells were detected in negative control groups which were without high temperature treatment. (**g**) Microscopic analysis of *K. pneumoniae* ATCC 43816 in response to phage Kp11 infection: (Upper Panel) Untreated bacteria were lysed by phages before cell division could occur, whereas (Lower Panel) heat-treated bacteria were able to undergo cell division despite exposure to the lytic phage. Phage Kp11 was added at time 0; scale bar is 2 μm; Black arrows indicate example cells undergoing lysis or division. Quantitative analysis further demonstrated a significantly higher fraction of dividing cells in the heat-treated group under phage pressure compared to untreated bacteria.

*Figure 2 continued on next page*

*Figure 2 continued*

The online version of this article includes the following video, source data, and figure supplement(s) for figure 2:

**Source data 1.** Data used for graphs presented in *Figure 2a–f*.

**Figure supplement 1.** Heat-induced phage tolerance was impacted by treatment time and temperature.

**Figure supplement 1—source data 1.** Data used for graphs presented in *Figure 2—figure supplement 1a and b*.

**Figure 2—video 1.** Untreated bacteria were lysed by phages before cell division could occur.

https://elifesciences.org/articles/105703/figures#fig2video1

**Figure 2—video 2.** Heat-treated bacteria were able to undergo cell division despite exposure to the lytic phage.

https://elifesciences.org/articles/105703/figures#fig2video2

## Heat-induced phage tolerance challenges phage therapy

We next assessed the infection of *K. pneumoniae* by phages Kp7, Kp9, Kp10, and Kp11 (*Huang et al., 2024*), as well as *E. coli* by phages T1 and T4, following pre-heat treatment. Our results revealed a consistent tolerance response across these experiments (*Figure 3a*, *Figure 3—figure supplement 1a and b*). This observation was further validated using clinical *K. pneumoniae* strains K359 and K381, both of which exhibited a marked reduction in phage sensitivity post-heat treatment (*Figure 3b*, *Figure 3—figure supplement 1c*).

Phage therapy often involves combining multiple phages to minimize the risk of resistance evolution (*Chan and Abedon, 2012*; *León and Bastías, 2015*). In light of this, we extended our investigation to phage combinations and found that heat-treated bacteria exhibited reduced sensitivity even to combined phage treatments (*Figure 3d*). Notably, we identified a Kp7-resistant strain, $R^{Kp7}$, which remained susceptible to Kp11 (*Figure 3—figure supplement 2a*) and was also sensitive to the combination of Kp7 and Kp11 (*Figure 3—figure supplement 2b*). This approved the effectiveness of combination therapy against spontaneous resistance-conferring mutations to a single phage (*León and Bastías, 2015*). Intriguingly, however, post heat treatment, the combination had a much higher propensity to fail due to the evolution of resistance to both phages (*Figure 3d*). These findings highlight the necessity of considering phage tolerance as a crucial factor in the design of effective phage therapies.

## Reduced phage DNA entry and altered envelope stress response in heat-treated bacteria

To investigate the mechanisms underlying phage tolerance in heat-treated bacteria, we first focused on the early stages of phage infection: adsorption and DNA injection. Our analysis revealed that heat treatment did not alter the phage adsorption rate (*Figure 4a*). Additionally, the production and structure of the bacterial capsule, the primary receptor for phage Kp11, remained unchanged (*Figure 4—figure supplement 1*). However, we observed a significant impairment in the entry of phage DNA into heat-treated bacteria (*Figure 4b*), suggesting a hindrance in the DNA injection process.

To explore this further, we employed liquid chromatography-mass spectrometry (LC-MS) for total protein analysis and identified that the down-regulated proteins were predominantly associated with the bacterial envelope (*Figure 4c and d*). In addition to a primary receptor, phages typically require a secondary receptor, often an outer membrane component (*Sueki et al., 2020*; *Sun et al., 2006*), to establish adsorption and initiate DNA injection. Based on these findings, we hypothesized that heat treatment may induce the loss or modification of certain envelope components, thereby inhibiting phage DNA injection and contributing to heat-induced phage tolerance.

The bacterial envelope serves as a barrier against environmental stresses (*Dehinwal et al., 2024*; *Mitchell and Silhavy, 2019*; *Silhavy et al., 2010*). We noticed that heat stress influenced not only the components mediating phage infection but also broader bacterial envelope dynamics, suggesting that overall envelope stress responses could be altered (*Eberlein et al., 2018*). To investigate this, we examined the response of heat-treated bacteria to various envelope stresses, including unfavorable conditions such as pH changes and exposure to toxic compounds (metals, bile salts, and polymyxins), and found substantial changes in susceptibility (*Figure 4e*). Interestingly, heat-treated bacteria exhibited increased survival against polymyxin, similar to phage tolerance (*Figure 4e*). Notably, heat-treated bacteria showed significantly reduced lipopolysaccharide (LPS) production, known to mediate

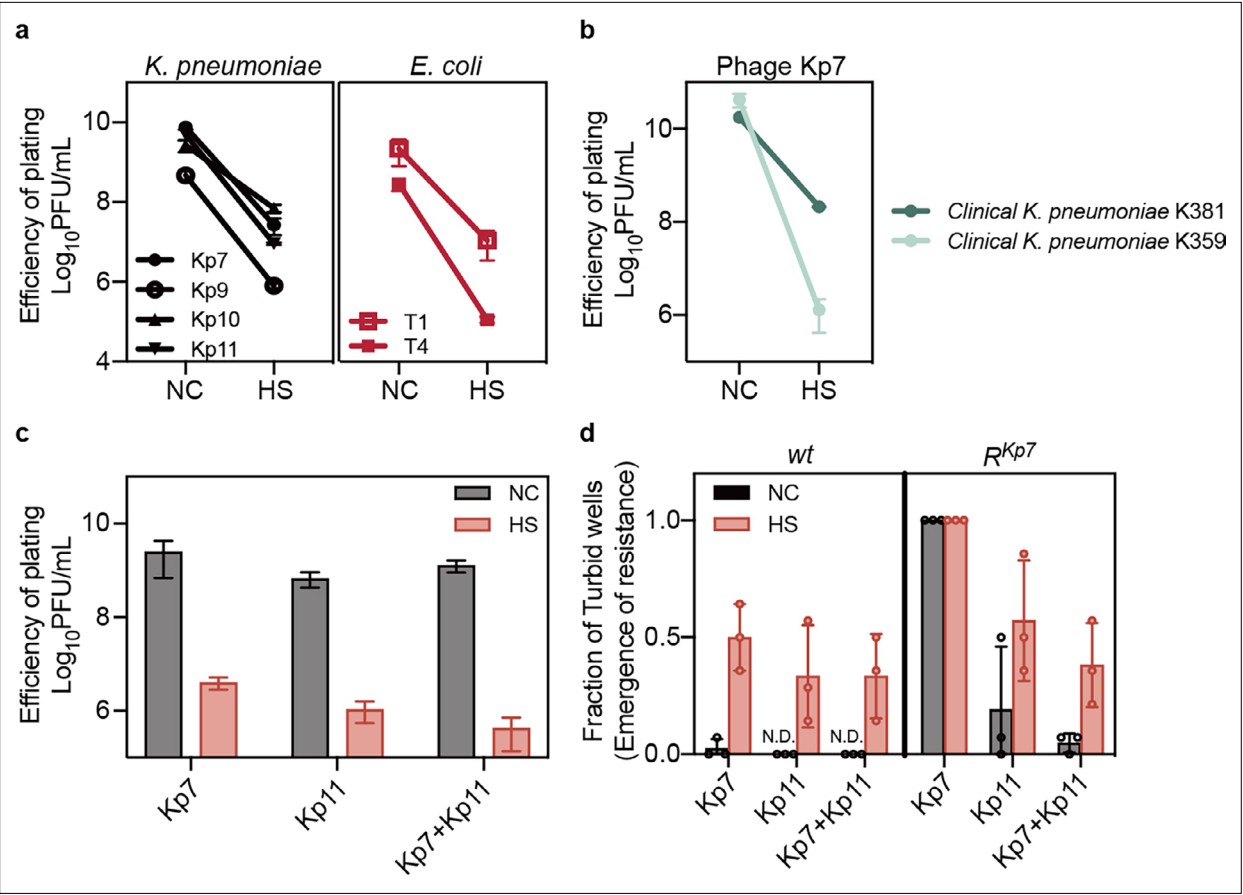

**Figure 3.** Heat-induced phage tolerance challenges phage therapy. (**a**, **b**) Small drop plaque assays were conducted using 10-fold serial dilutions of phages Kp7, Kp9, Kp10, or Kp11 against *K. pneumoniae* ATCC 43816, as well as phage Kp7 against clinical *K. pneumoniae* strains K357 and K381, and phages T1 or T4 against *E. coli KLY*, both with and without heat treatment. Data are presented as mean ±s.d. of three independent replicates. (**c**) Efficiency of plating assays was performed for individual phages or phage combination on *K. pneumoniae* ATCC 43816, with and without heat treatment. Data are presented as mean ±s.d. of three independent replicates. (**d**) Bacteria (~10³ CFU) with and without heat treatment were cultured with either a single phage or phage combination at an MOI of 100 for 24 hr. The fraction of turbid wells indicates resistance emergence. Data are presented as mean ±s.d. of three independent replicates and 14 wells for each replicate.

The online version of this article includes the following source data and figure supplement(s) for figure 3:

**Source data 1.** Data used for graphs presented in *Figure 3a–d*.

**Figure supplement 1.** Phage tolerance induced by heat stress is a consistent response observed across various bacterial strains.

**Figure supplement 2.** Heat stress–induced phage tolerance reduces the efficacy of phage combinations.

**Figure supplement 2—source data 1.** Data used for graphs presented in *Figure 3—figure supplement 2a and b*.

polymyxin adsorption (*Yin et al., 2020*; *Figure 4f*). Further experiments using the LPS inhibitor CHIR-090 confirmed that bacterial survival under polymyxin treatment increased with LPS reduction (*Figure 4g*). Given that polymyxin is a last-resort antibiotic for treating carbapenem-resistant *K. pneumoniae*, this cross-tolerance with phages might present a significant challenge to already scarce treatment options.

### Phage resistance in heat-evolved *pspA* mutant

Heat treatment enhances bacterial tolerance to lytic phages, prompting us to investigate whether mutants surviving heat treatment could also withstand subsequent phage infections. To address this, we conducted a serial evolutionary experiment under selective heat stress (*Figure 5a*). In line with our hypothesis, we identified two distinct mutants, BC2G11C1 and BC3G11C2, which exhibited significantly enhanced survival following heat treatment (*Figure 5b*). However, only BC2G11C1 showed significantly reduced susceptibility to phage infection (*Figure 5c*). Moreover, this strain maintained

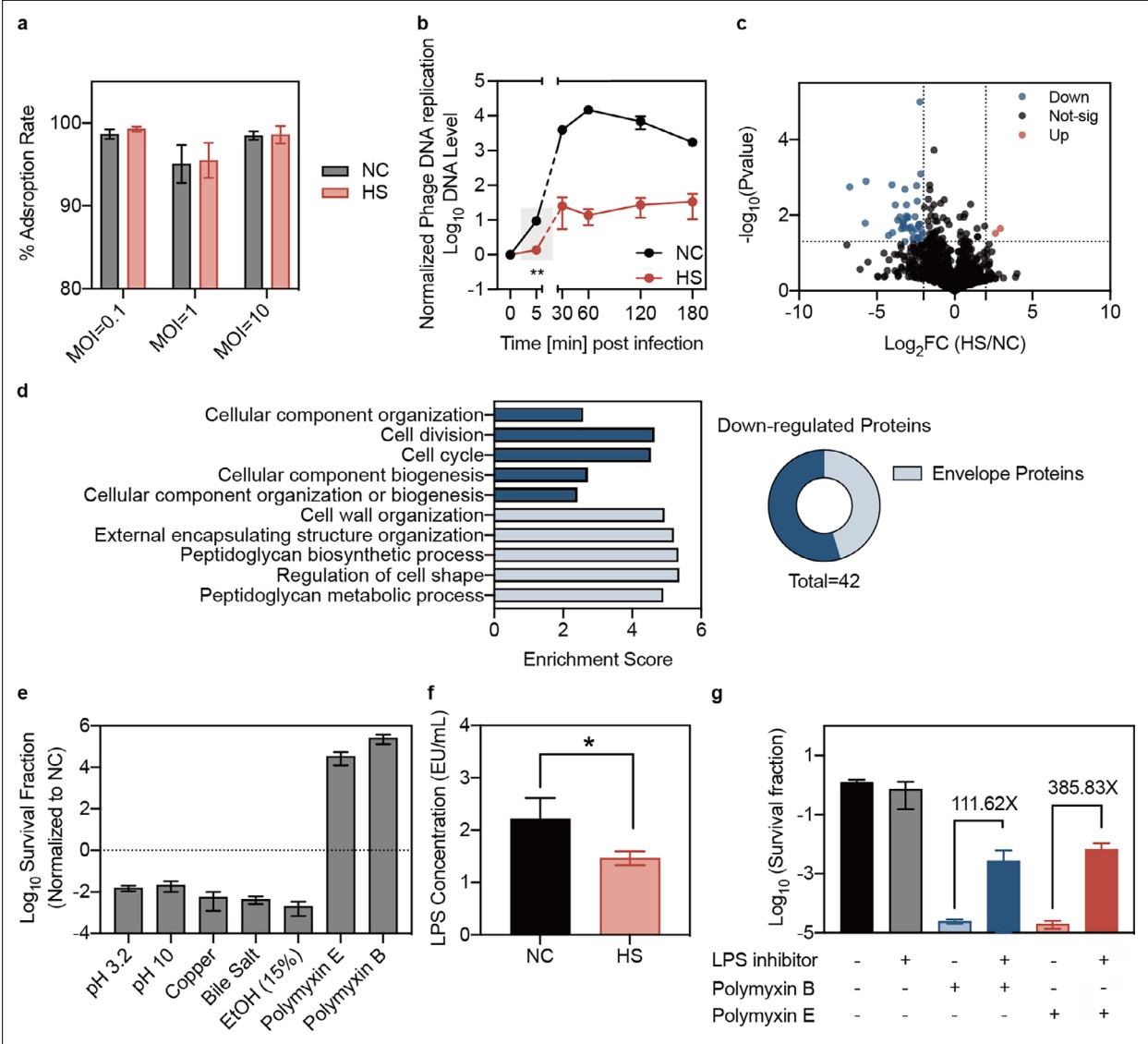

**Figure 4.** Reduced phage DNA entry and altered envelope stress responses in heat-treated bacteria. (**a**) Adsorption assays of phage Kp11 to *K. pneumoniae* ATCC43816 were performed with and without heat treatment at various MOIs of 0.1, 1, and 10 for 3 min. Data are presented as mean ±s.d. of three independent replicates. (**b**) The phage DNA levels in bacteria were measured at different time points. Phage Kp11 (MOI ~100) was added at time 0. Data are presented as mean ±s.d. of three biological replicates. Statistical significance was determined using significance levels set at **p<0.01 (p=0.0046). (**c**) Comparison of the protein profiling results for *K. pneumoniae* ATCC43816 with and without heat treatment. The color of each dot represents a p value calculated based on the $\log_2$ ratio of protein abundance in the two populations ($\log_2$(HS/NC)) (see Materials and methods for the statistical test used). The best p values from all technical replicates are shown. The intensity is the normalized mean mass spectrometry peak intensity of each protein associated with the best p value among the three replicates. (**d**) Gene ontology (GO) analysis of the downregulated proteins in biological process. (**e**) Normalized survival fraction of heat-treated *K. pneumoniae* ATCC43816 under various stresses, including pH changes (pH 3.2 and pH 10 for 1 hr), copper (5 mM $CuSO_4$ for 1 hr), ethanol (15% ethanol for 1 hr), bile salt (10 mg/mL for 3 hr) and polymyxin (Polymyxin B, E 25 µg/mL for 1 hr). Data are presented as mean ±s.d. of three independent replicates. (**f**) The production of LPS in bacteria was quantified using the Chromogenic LAL Endotoxin Assay. Bars show mean ± s.d. of three biological replicates. Statistical significance was determined using significance levels set at *p<0.05 (p=0.0377). (**g**) Survival fraction of *K. pneumoniae* treated with polymyxin for 1 hr, with or without the LPS inhibitor CHIR-090. Data are presented as mean ±s.d. of three independent replicates.

The online version of this article includes the following source data and figure supplement(s) for figure 4:

**Source data 1.** Data used for graphs presented in *Figure 4a, b* and *Figure 4d–g*.

**Figure supplement 1.** Production and structure of the bacterial capsule in heat-treated and untreated bacteria.

**Figure supplement 1—source data 1.** Data used for graphs presented in *Figure 4—figure supplement 1a*.

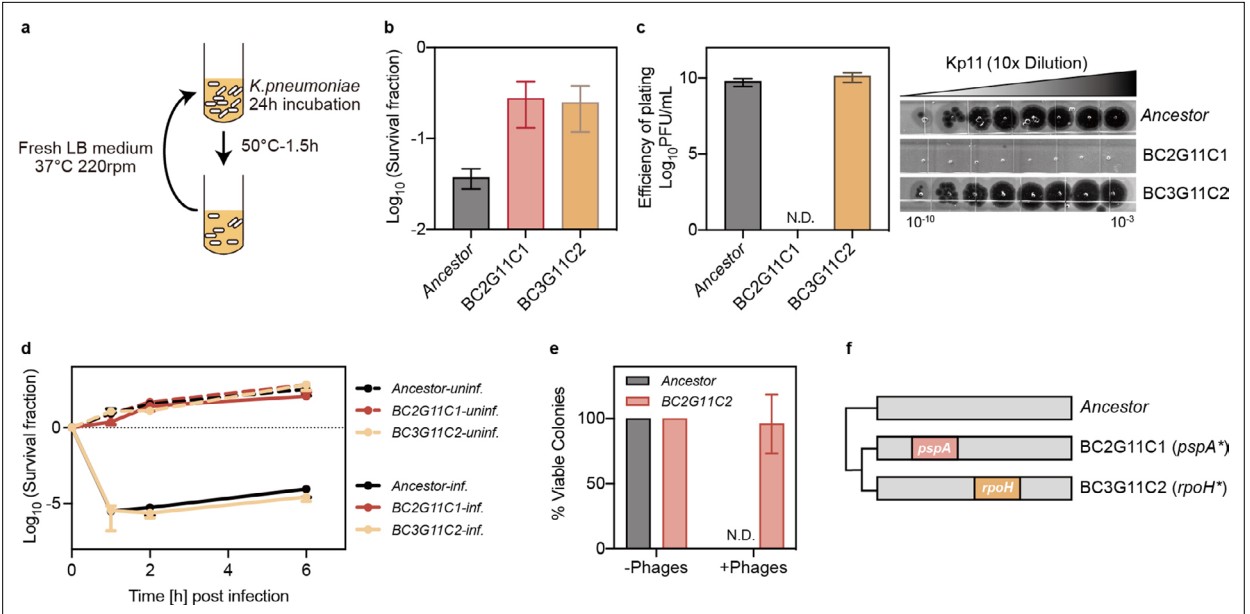

**Figure 5.** Mutation in *pspA* confers dual insensitivity to thermal and phage stress. (**a**) Experimental design for cyclic exposure to heat treatment, where cultures of ATCC 43816 were subjected to a 50 °C bath for 90 min. Following heat treatment, cultures were resuspended in fresh medium and incubated for 24 hr. (**b**) Survival fraction of ancestral and evolved strains at 50 °C for 90 min. Data are presented as mean ± s.d. of three independent replicates. BC: Batch Culture, G: Evolution Cycle Number, C: Colony. BC2G11C1 refers to the first colony from batch culture 2 after 11 rounds of heat treatment. (**c**) Efficiency of plating assay for phage Kp11 to ancestral and evolved strains. Data are presented as mean ± s.d. of three independent replicates. (**d**) Survival fraction of ancestral and evolved strains from phage Kp11 infection (MOI ~100). Curves show the mean ± s.d. of three independent replicates. (**e**) Viable colonies of ancestral and evolved strains grown on both phage plates ($10^8$ PFU) and phage-free plates. Data are presented as the mean ± s.d. of three independent replicates. (**f**) Nonsynonymous mutations identified in the strains isolated from the evolved culture.

The online version of this article includes the following source data and figure supplement(s) for figure 5:

**Source data 1.** Data used for graphs presented in *Figure 5b–e*.

**Figure supplement 1.** Impact of *pspA* mutations on bacterial survival and phage resistance.

growth under phage treatment, indicating that the mutation conferred phage resistance (*Figure 5d and e*). Thus, the BC2G11C1 mutant acquired both phage resistance and heat tolerance.

Whole-genome sequencing of the evolved strains revealed a truncation in the *pspA* gene in the phage-resistant mutant, which was responsible for the reduced sensitivity to both heat and phage treatment. In contrast, the other strain harbored a mutation in the *rpoH* gene, which conferred heat tolerance but did not affect phage resistance (*Figure 5f*). The absence of similar phenotypes in the *pspA* knockout strain (*Figure 5—figure supplement 1*) suggested that this was a gain-of-function mutation in the *pspA* gene. This was further supported by the failure to restore the phenotypes upon reintroduction of the wild type *pspA* gene into the mutant strain; however, overexpression of the mutated *pspA* gene in the ancestral strain exhibited a similar phenotype (*Figure 5—figure supplement 1*).

Phage shock proteins, encoded by the *psp* operon, are reported to be induced by both heat stress and phage infection (*Flores-Kim and Darwin, 2016*; *Model et al., 1997*). However, the underlying mechanism has remained elusive. Our findings led us to hypothesize that the *psp* operon might function through envelope biogenesis, thereby linking responses to heat and phage treatment.

To test our hypothesis, we first investigated whether the phage life cycle in the *pspA\** mutant was disrupted at an early stage, building on previous results regarding heat-induced phage tolerance. Adsorption assays revealed a significant impairment in phage adsorption to *pspA\**, suggesting alterations in receptor availability (*Figure 6a*). Further analysis corroborated this observation, demonstrating a reduction in capsular production (*Figure 6b*) and a complete loss of capsular structure (*Figure 6c*).

Given the critical role of *K. pneumoniae* capsules in evading host immunity during septicemia (*Huang et al., 2022*), we assessed the immune evasion capacity of the *pspA\** mutant. The results

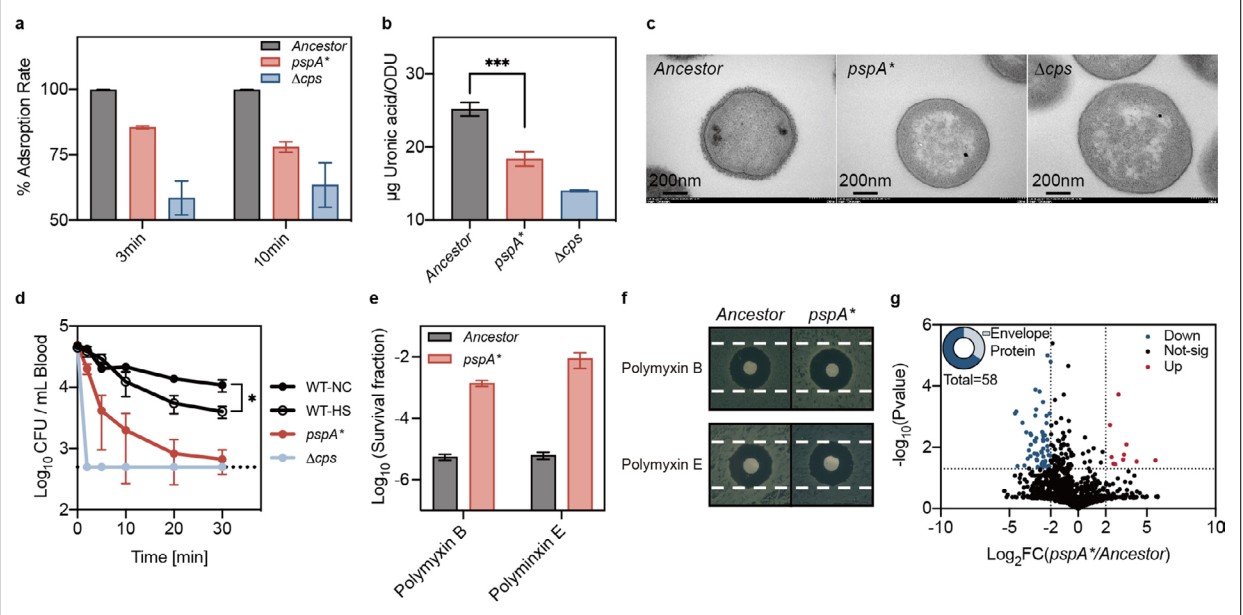

**Figure 6.** *pspA* mutation impaired phage adsorption and altered envelope stress responses. (**a**) Adsorption assay of phage Kp11 to the ancestral strain *K. pneumoniae* ATCC 43816, *pspA\** and the non-capsular strain *Δcps* at an MOI of 0.1. (**b**) Quantification of capsular production of the ancestral strain, *pspA\** and *Δcps*. The capsular polysaccharide (CPS) was quantified using the uronic acid method, presented as the amount of uronic acid per OD600. Data are presented as the mean ±s.d. of three independent replicates; statistical significance was assessed using an unpaired two-tailed t-test, with statistical significance levels set at ***p<0.005 (p=0.001). (**c**) Detection of bacterial capsular structure using transmission electron microscopy (TEM). (**d**) Virulence traits were assessed by intravenously (i.v.) infection of ICR mice with ~10⁶ of each bacterial strain (ancestral with and without heat treatment, *pspA\**, and *Δcps*) to determine bacteremia levels. Curves show the mean of three independent replicates, and error bars are s.d. of the mean. Statistical significance was assessed using an unpaired two-tailed t-test, with significance levels set at *p<0.05 (p=0.011). (**e**) Survival fraction of bacteria (ancestral and evolved strain *pspA\**) after polymyxin treatment (polymyxin B and E, 25 µg/mL for 1 hr). Data are presented as the mean ± s.d. of three independent replicates. (**f**) Disk fusion assay (polymyxin B and E, 10 µg) of the ancestral and evolved strain *pspA\**. (**g**) Comparison for the protein profiling results for the *ancestor* and *pspA\**. The color of each dot represents a p value calculated based on the log₂ ratio of protein abundance in the two populations (log₂(*pspA\*/Ancestor*)).

The online version of this article includes the following source data and figure supplement(s) for figure 6:

**Source data 1.** Data used for graphs presented in *Figure 6a, b* and *Figure 6d, e*.

**Figure supplement 1.** Superior protection of untreated ancestral strain ATCC 43816 against immune clearance of septic infection.

**Figure supplement 1—source data 1.** Data used for graphs presented in *Figure 6—figure supplement 1a–c*.

showed that *pspA\** was rapidly cleared from mouse blood, demonstrating a significant reduction in immune evasion capacity (*Figure 6d, Figure 6—figure supplement 1*). A mild reduction in immune evasion was also observed in heat-treated bacteria (*Figure 6d*). Notably, the *pspA\** mutant exhibited increased survival to polymyxin treatment (*Figure 6e*) without a corresponding increase in the minimum inhibitory concentration (MIC) (*Figure 6f*), suggesting a tolerant phenotype mediated by a reduction in LPS. Mass spectrometry analysis indicated that many down-regulated proteins in *pspA\** were envelope-associated (*Figure 6g*), implying an important role for the *pspA* gene in maintaining envelope integrity.

## Discussion

Previous reports have demonstrated that antibiotic persistence confers a distinct advantage to lysogenic phages but does not protect bacteria from lytic phages (*Pearl et al., 2008*). Here, we identified bacterial persistence against lytic phages, where bacteria survived extended phage treatment without developing resistant mutations (*Figure 7*). This introduces a novel strategy for bacterial defense against phages. Notably, our research revealed that heat stress enhances phage tolerance, subsequently promoting the emergence of resistance, aligned with observations in antibiotic research (*Levin-Reisman et al., 2019*; *Levin-Reisman et al., 2017*). This parallel emphasizes the necessity of

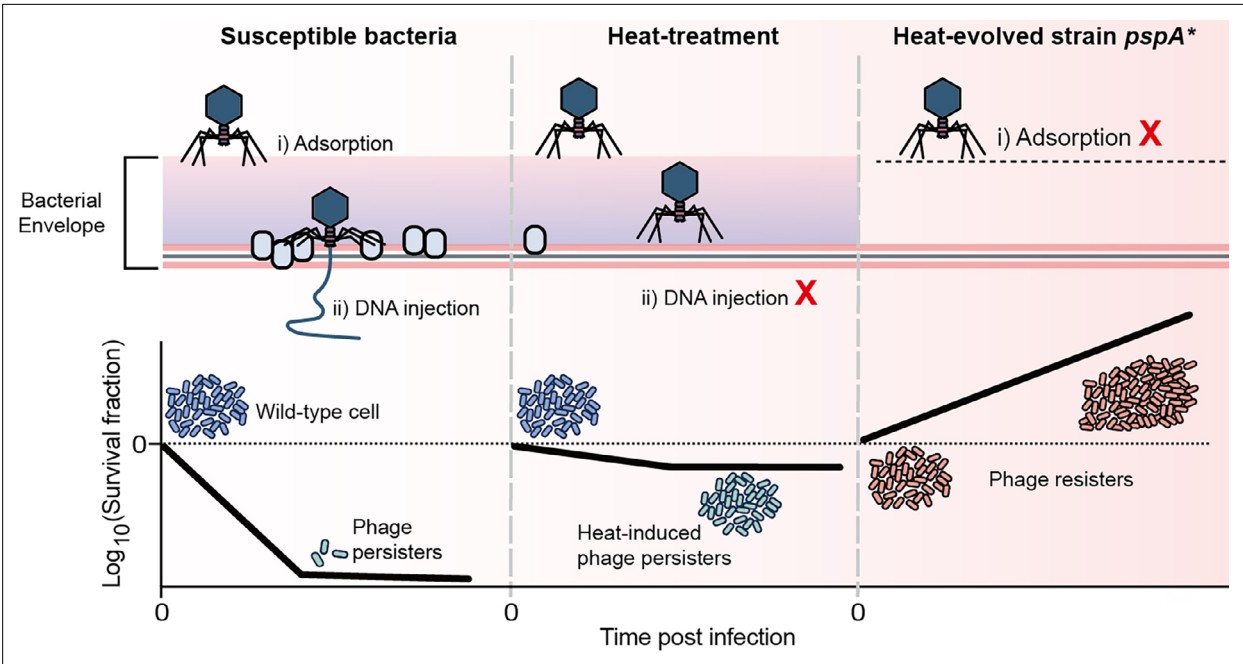

**Figure 7.** Schematical model of this work. A small proportion of phage-sensitive bacteria can survive during lytic phage infections, exhibiting persistence against phage. Heat treatment systematically reduces envelope-associated proteins in bacteria, inhibiting the entry of phage DNA and thereby inducing phage tolerance. Mutation in the gene pspA resulted in the loss of capsule, conferring receptor-deficient phage resistance.

cautious and strategic design of phage therapy. In line with recent work by *Fernández-García et al., 2024*, which showed that lytic phages can induce antibiotic persistence, our findings further highlight the existence of phage persistence, adding depth to bacterial survival strategies and reinforcing the complexity of phage therapy.

Heat, as a universal stress, induces protein misfolding and aggregation, despite the activation of heat shock proteins that mitigate these effects (*Feder and Hofmann, 1999*; *Mogk et al., 2019*). The accumulation of misfolded envelope proteins can become toxic, leading to cell death (*Balchin et al., 2016*; *Konovalova et al., 2016*; *Takai et al., 1998*). Correspondingly, the relatively low efficiency of translocation and assembly of envelope proteins (*Bakkes et al., 2010*; *Dalbey and Kuhn, 2012*; *Vigh et al., 2007*) renders them more vulnerable under heat treatment (*Schiffrin et al., 2017*), resulting in a decrease in abundance. This reduction in envelope protein might be an adaptive response to elevated temperatures, as evidenced by the observations following heat treatment. Consequently, this would lead to a decrease in phage receptors (such as capsule and outer membrane protein FhuA) and the receptor for polymyxin (LPS), thereby conferring tolerance. Recent studies have shown that heat shock potentiates aminoglycosides against Gram-negative bacteria by enhancing antibiotic uptake and promoting protein aggregation (*Lv et al., 2023*), indicating that heat stress can contribute to bacterial sensitivity to antibiotics, which could also extend to phages in our study.

This phenomenon may also occur in the heat-tolerant mutant *pspA\**, where the reduction in envelope components resulted in insusceptibility to phages and polymyxins, highlighting a potential regulatory balance governed by PspA. PspA, a key regulator in the phage shock protein system, functions as part of the envelope stress response system in bacteria, preventing membrane depolarization and ensuring the envelope stability (*Flores-Kim and Darwin, 2016*; *Horstman and Darwin, 2012*; *Kleerebezem and Tommassen, 1993*; *Yamaguchi et al., 2010*). Recent studies have shown that PspA is analogous to ESCRT-III, a component of the ESCRT machinery in eukaryotic cells, which plays a crucial role in membrane remodeling (*Junglas et al., 2021*; *Liu et al., 2021*).

The bacterial envelope acts both as a defense barrier against external stresses and as a conduit for invaders. While it reduces the invasion of harmful substances and maintains intra-extracellular balance (*Arts et al., 2015*; *Raivio, 2014*; *Wood, 2015*), it also provides contacts for invaders; for example, capsules serve as receptors for phages (*Latka et al., 2021*), and LPS interacts with polymyxins (*Manioglu et al., 2022*). Thus, heat-induced alteration to the envelope could be a double-edged

sword. Clinically, heat stress would not only impact phage susceptibility but also alter the interaction between *K. pneumoniae* and the host immune system. This dual impact underscores the complexity of bacterial adaptive responses and suggests that environmental factors can significantly influence the outcome of phage therapy. Furthermore, increased sensitivity to acidic conditions and bile salts induced by heat stress challenges bacterial survival, particularly relevant to *K. pneumoniae* colonization sites in the human body (*Gasink et al., 2009*; *Martin et al., 2016*).

Lastly, the envelope stress responses induced by heat in bacteria lead us to consider the role of fever. We found that heat-induced phage tolerance is both time- and temperature-dependent; even moderately elevated temperatures, such as 42 °C, can induce phage tolerance with prolonged treatment, indicating alterations in the bacterial envelope. This suggests that elevated body temperature, akin to fever, may affect bacterial survival and colonization by modulating their responses to the host immune system and antimicrobial agents. Since elevated temperatures also alter life-history traits in phage communities (*Greenrod et al., 2024*), they may similarly impact bacterial responses to heat stress, phage, and antibiotic treatments, underscoring the need for further research into the role of fever in bacterial growth and survival during therapy.

## Materials and methods

### Bacteria, phages, and plasmids

*K. pneumoniae* and *E. coli* strains used in this work are listed in *Supplementary file 1*. *E. coli* strain *DH5α* was used for DNA cloning (*Seidman and Struhl, 2001*). The phages used in this work are also listed in *Supplementary file 1*. Plasmid constructions were performed in *E. coli* strain *DH5α* using standard methodologies and are listed in *Supplementary file 2*.

### Phage purification and concentration

Overnight cultured bacteria were inoculated into 200 mL of fresh LB medium and grown to an $OD_{600}$ of ~0.3. Then, 100 µL of phage stock was added to the cultures for a few hours, after which the cultures were centrifuged at 4 °C, at 4000 rpm when cultures turned to transparency. The supernatants were filtered using a 0.22 µm filter to wipe off bacteria, and the phage cultures were concentrated using ultracentrifuge tubes (100 KD). Filtered SM buffer was added to the concentrated phage cultures until a stable volume was achieved, and the final phage stocks were stored at 4 °C (*Bonilla et al., 2016*).

### Killing dynamic of phage infection and phage persisters definition

Killing dynamic of phage infection to *K. pneumoniae* was performed by treating cells (~$10^6$ CFU/mL) with lytic phage Kp11 (MOI ~100) for 120 min while shaking at 220 rpm in a 15 mL tube. Samples were taken at indicated time points, washed three times with PBS buffer, and CFUs were evaluated by plating on LB agar plates before and after phage infection. To determine the persistence fraction and obtain phage persisters, a single colony from post-phage treatment was inoculated into fresh LB broth and grown at 37 °C with shaking at 220 rpm for 6 hr. Then each culture was collected and dropped on solid LB agar plates supplemented with and without phages. Phage persisters were identified as those solely capable of growth on phage-free plates, with the fraction of persisters estimated by subtracting the number of resistant mutants from the total CFUs. After identification, descendants of persisters were diluted to ~$10^6$ CFU/mL and infected with phage Kp11 for 40 minutes, with CFUs evaluated by plating before and after phage treatment over several cycles.

### Plaque assays

Phage titers and efficiency of plating were determined using the small drop plaque assay method (*Goldfarb et al., 2015*). 200 µL of bacterial culture was mixed with 5 mL of melted 0.7% soft LB agar medium and overlaid on 1.5% solid LB plates to form bacterial lawn. After allowing the bacterial lawn to dry for 15 min, 5 µL of 10-fold serial dilutions of phages in SM buffer were dropped to the lawn. Once the drops had dried, the plates were inverted and incubated at 37 °C overnight. Plaque-forming units (PFUs) were determined by counting the derived plaques after incubation, with lysate titers calculated as PFUs per mL. If no individual plaques could be observed, a faint lysis zone across the drop area was considered equivalent to 10 plaques. The efficiency of plating (EOP) was measured by comparing results from control and treated bacteria.

## ScanLag analysis

Bacteria were diluted serially and plated on solid LB agar medium. The plates were placed in a ScanLag (*Levin-Reisman et al., 2010*) setup at 37 °C, which consists of an array of office scanners that automatically capture images of the plates every 20 min, thus monitoring the appearance of colonies. An automated image analysis application extracts the distribution of appearance time of the colonies.

## Growth dynamic of phage infection culture

In this work, bacterial cultures with and without heat treatment were diluted to $10^5$–$10^6$ CFU/mL in LB medium and infected with high concentrations of phages at an MOI of 100. Optical density measurements at a wavelength of 600 nm were taken every 10 min using a TECAN Infinite 200 plate reader in a 96-well plate.

## Fluctuation test for measuring mutation rate (modified from Delbrück-Luria Fluctuation test)

To measure the mutation rate of heat-treated bacteria to phage, several replicas of culture (~1000 bacteria per well) were grown overnight in a 96-well plate to $10^9$ CFU/mL in 200 µL and treated with and without heat stress. Then each culture was plated onto solid agar plates supplemented with $10^9$ PFUs of phages and incubated at 37 °C. The fraction of empty plates was used to calculate the mutation rate from five biological repeats of 14 replicas (*Luria and Delbrück, 1943*).

## Resistance emergence fraction experiments

Bacteria culture was seeded in a 96-well plate to achieve an approximate total bacterial load of 1000 CFU per well, minimizing the impact of spontaneous resisters and ensuring that any observed resistance was due to division during phage infection. Bacteria in each well were incubated with phages at various MOIs for 24 hr. Given the exceptionally low total bacterial load utilized in this experiment, the potential for phage-resistant bacteria due to stochastic mutations would be minimized. The high MOI guaranteed that each bacterium was likely to be infected by at least one phage particle. Due to the rapid infection and lysis in this system, the probability of observing turbid wells after 24 hr was minimal. The fraction of turbid wells was used to calculate the phage resistance emergence from three biological repeats of 14 replicates.

## Time-lapse microscopy and image acquisition

To observe bacterial dynamics under phage treatment, time-lapse microscopy was performed using an FV3000 with a temperature-controlled stage. *K. pneumoniae* cultures with and without heat treatment were exposed to lytic phage Kp11 at an MOI of 100. For time-lapse imaging, bacterial cultures were transferred to a 35 mm glass-bottom dish (MatTek Corporation) and incubated at 37 °C in a humidified atmosphere. The microscope was set to capture images at regular intervals (e.g. every 2 min for untreated bacteria and 10 min for heat-treated bacteria). A 100×oil immersion objective was used to obtain high-resolution images. Images were captured using Ti2-E (Nikon, Japan) with a live-cell system FV3000 RS (Olympus, Japan). The resulting image stacks were analyzed using ImageJ.

## Phage adsorption assay

The indicated bacterial cultures were diluted to fresh LB medium and infected with phages at different MOIs (0.1, 1, 10). 500 µL of samples were collected at 3- and 10 min post-infection and centrifuged for 1 min at 12,000 rpm. 500 µL of LB broth with 50 µL of chloroform were added to the supernatant, which was then placed on ice for 10 min to remove bacterial body. Phage adsorption to the indicated bacteria was measured by titrating the free phages present in supernatant (*Kim et al., 2019*).

## Transmission electron microscopy

The samples were fixed in a mixture of 2% paraformaldehyde and 2.5% glutaraldehyde, followed by four washes with 0.1 M phosphate buffer (PB). Post-fixation was performed using 1% osmium tetroxide and 1.5% tetrapotassium hexacyanoferrate trihydrate for 1 hr at 23 °C. Samples were then dehydrated through graded ethanol solutions (50%, 70%, 80%, 90%, 100%, 100%, 100%) for 15 min each. Next, they were infiltrated twice with 1,2-epoxypropane for 15 min each, followed by gradient infiltration with a mixture of 1,2-epoxypropane and Epon 812 resin for 8 hr (SPI, America). The samples

were then embedded in pure Epon 812 twice and polymerized in an oven at 60 °C. Blocks of polymerized resin were sectioned using a Leica EM UC7 ultramicrotome (Wetzlar, Germany), yielding ultrathin sections (70 nm) mounted on coated copper grids. Sections were stained on-grid with 2% uranyl acetate for 25 min and lead citrate for 5 min. Imaging was conducted using an H-7650B transmission electron microscope (Hitachi, Tokyo, Japan).

## Phage DNA injection and replication kinetics monitoring

Indicated bacteria were infected with phages at an MOI of 10. At indicated time points, cells were harvested, and total DNA (including both bacterial and phage DNA) was extracted. qPCR was performed using primers specific to the phage genome (shown in *Supplementary file 2*), allowing quantification of phage DNA levels by comparing qPCR results.

## Mass spectrometry

10 mL of bacteria for each sample were used for total protein extraction by using Bacterial Protein Extraction Kit (C600596, Sangon) and subjected to liquid chromatography-mass spectrometry (LC-MS). Briefly, proteins were disulfide reduced with 25 mM dithiothreitol (DTT) and alkylated with 55 mM iodoacetamide. In-gel digestion was performed using sequencing-grade modified trypsin in 50 mM ammonium bicarbonate at 37 °C overnight. The peptides were extracted twice with 1% trifluoroacetic acid in 50% acetonitrile for 30 min. The peptide extracts were then centrifuged using a SpeedVac. For LC-MS/MS analysis, peptides were separated by a 60-min gradient elution at a flow rate of 0.300 µL/min with a Thermo-Dionex Ultimate 3000 HPLC system, directly interfaced with the Thermo Orbitrap Fusion mass spectrometer. The analytical column was a homemade fused silica capillary column (75 µm ID, 150 mm length; Upchurch, Oak Harbor, WA) packed with C-18 resin (300 A, 5 µm; Varian, Lexington, MA). Mobile phase A consisted of 0.1% formic acid, and mobile phase B consisted of 100% acetonitrile and 0.1% formic acid. The Orbitrap Fusion mass spectrometer was operated in the data-dependent acquisition mode using Xcalibur 3.0 software, and there is a single full-scan mass spectrum in the Orbitrap (350–1550 m/z, 120,000 resolution) followed by 3 s data-dependent MS/MS scans in an Ion Routing Multipole at 30% normalized collision energy (HCD). The MS/MS spectra from each LC-MS/MS run were searched against the selected database using the Proteome Discovery searching algorithm (version 1.4).

## Evolutionary protocol of cyclic exposure to heat stress

The cyclic evolution protocol (*Figure 5*) consists of two steps: killing by heat stress at 50 °C for 90 min and 24 hr growth. Three replicate populations evolved, each initiated from a single colony isolated from the ancestral glycerol stock in 2 mL of fresh LB medium, which was shaken at 220 rpm at 37 °C for 24 hr. The cultures were transferred to an elevated temperature of 50 °C for 90 min. Then the cultures were resuspended in 2 mL of fresh LB medium and grown for 24 hr at 37 °C with shaking. Half of the regrown cultures were used for analysis. After each cycle, samples from each culture were diluted serially and plated on solid LB agar medium to determine the survival.

## Mouse infection

Bacteremia modeling in mice followed the approach described by *Huang et al., 2022*. Female ICR mice (6–8 weeks old) were intravenously (i.v.) injected with *K. pneumoniae*. Bacteria were collected from pre-counted frozen stocks and resuspended in Ringer's solution to the desired concentration for injection (100 µL). To calculate actual infectious doses, bacterial CFUs were plated 1 day prior to infection. Bacterial levels in the blood were measured by orbital hemorrhage at set time points post-infection, with the initial bacterial load (0 min) determined by dividing inoculum CFUs by blood volume (0.07 mL/g of body weight). Bacterial counts in organs were measured similarly, using tissue homogenates and estimating residual blood volume at 0.1 mL/g of organ. The total bacterial burden was calculated by summing the CFU counts in the blood and organs, and the survival rate was determined by dividing the total surviving bacteria by the initial inoculum CFUs.

## Genomic extraction and whole-genome sequencing

Genomic DNA was extracted using the Tianamp Bacteria DNA Kit (Tiangen). Sequencing was performed by Sangon using the Illumina-HiSeq with 150 bp paired-end reads, at an average coverage

of 100–200. Identification of single nucleotide polymorphisms (SNPs) was performed by aligning the raw Illumina reads to the genome of the reference strain ATCC 43816 using bwa version 0.7.18.

## Statistical analysis

All experiments were performed with at least three independent biological replicates unless otherwise specified. Bar graphs display the mean ± standard deviation (s.d.). Unpaired, two-tailed t-tests were used to calculate p values for comparisons of survival rates, EOP, and LPS production, with statistical significance set at $p<0.05$. For the protein profiling experiments, differential protein abundance between the heat-treated and control groups was analyzed using Student's t-tests for each protein. Multiple comparisons were adjusted using the Benjamini-Hochberg false discovery rate (FDR) correction to control for Type I errors. Proteins with a corrected p-value $<0.05$ were considered significantly differentially expressed.

## Acknowledgements

We thank Quan-Jiang Ji (ShanghaiTech University) for providing plasmids used in K pneumoniae genome editing and Nathalia. Balaban Q (Hebrew University of Jerusalem) for the *E. coli* strains KLY and metG*. We are grateful to Guan-Xiang Liang, Yao-Yu Yang, and Gao-Pu Zhang (Tsinghua University) for the phages, and Jun-Hao Zhu (Institute of Microbiology, Chinese Academy of Sciences) for manuscript comments. We acknowledge Hai-Teng Deng and Xian-Bin Meng (Proteomics Facility, Technology Center for Protein Sciences, Tsinghua University) for assistance with protein MS analysis, and Jing-Jing Wang (Cell Biology Facility, Center for Biomedical Analysis, Tsinghua University) for technical support. We also thank the Cell Biology Facility (Center for Biomedical Analysis, Tsinghua University) for equipment support. National Key Research and Development Program of China (2022YFC2303202); Tsinghua-Peking Joint Center for Life Sciences (20111770319); Tsinghua University Dushi Program (20231080044)

## Additional information

### Funding

| Funder | Grant reference number | Author |
| --- | --- | --- |
| National Key Research and Development Program of China | 2022YFC2303202 | Jia-Feng Liu |
| Tsing-Pekin Joint Center for Life Sciences | 20111770319 | Jia-Feng Liu |
| Tsinghua University Dushi program | 20231080044 | Jia-Feng Liu |

The funders had no role in study design, data collection and interpretation, or the decision to submit the work for publication.

### Author contributions

Fan Zhang, Conceptualization, Data curation, Software, Formal analysis, Validation, Investigation, Visualization, Methodology, Writing – original draft, Writing – review and editing; Hao-Ze Chen, Investigation; Bo Zheng, Data curation, Software, Formal analysis; Liang Huang, Resources; Ye Xiang, Jing-Ren Zhang, Writing – review and editing; Jia-Feng Liu, Conceptualization, Resources, Supervision, Funding acquisition, Investigation, Visualization, Writing – original draft, Project administration, Writing – review and editing

### Author ORCIDs

Fan Zhang https://orcid.org/0009-0001-1425-8112
Hao-Ze Chen https://orcid.org/0009-0007-9269-3025
Bo Zheng https://orcid.org/0000-0003-3911-7217
Liang Huang https://orcid.org/0000-0001-8414-7883

Ye Xiang ⓘ https://orcid.org/0000-0003-0230-9522
Jing-Ren Zhang ⓘ https://orcid.org/0000-0003-4973-4243
Jia-Feng Liu ⓘ https://orcid.org/0000-0003-3301-7194

### Ethics

All mouse experiments were approved by the Institutional Animal Care and Use Committee in Tsinghua University under the protocol 23-ZJR1.

Reviewer #1 (Public review): https://doi.org/10.7554/eLife.105703.3.sa1
Reviewer #2 (Public review): https://doi.org/10.7554/eLife.105703.3.sa2
Author response https://doi.org/10.7554/eLife.105703.3.sa3

---

## Additional files

### Supplementary files

MDAR checklist

Supplementary file 1. List of bacterial strains and phages used in this study.

Supplementary file 2. List of plasmids and primers used in this study.

### Data availability

All data generated or analysed during this study are included in the manuscript and supporting files; source data files have been provided for figures.

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
