## [Editor Report · eLife Assessment]

This **important** study analyzes the effect of heat treatment on phage-bacterial interactions and **convincingly** shows that prior heat exposure alters the bacterial cell envelope, enhancing persistence and bacterial survival when exposed to lytic phages. The study will interest researchers working on antibiotic resistance, tolerance, and phage therapy.

---

## [Referee Report · Reviewer #1 (Public review)]

Summary:

In this interesting and original paper, the authors examine the effect that heat stress can have on the ability of bacterial cells to evade infection by lytic bacteriophages. Briefly, the authors show that heat stress increases tolerance of Klebsiella pneumoniae to infection by the lytic phage Kp11. They also argue that this increased tolerance facilitates the evolution of genetically encoded resistance to the phage. In addition, they show that heat can reduce the efficacy of phage therapy. Moreover, they define a likely mechanistic reason for both tolerance and genetically encoded resistance. Both lead to a reorganization of the bacterial cell envelope, which reduces the likelihood that phage can successfully inject their DNA.

Strengths:

I found large parts of this paper well written and clearly presented. I also found many of the experiments simple yet compelling. For example, the experiments described in figure 3 clearly show that prior heat exposure can affect the efficacy of phage therapy. In addition the experiments shown in figure 4 and 6 clearly demonstrate the likely mechanistic cause of this effect. The conceptual figure 7 is clear and illustrates the main ideas well. I think this paper would be publishable even without its central claim, namely that tolerance facilitates the evolution of resistance. The reason is that the effect of environmental stressors on stress tolerance has to my knowledge so far only been shown for drug tolerance, not for tolerance to an antagonistic species.

Weaknesses:

I did not detect any weaknesses that would require a major reorganization of the paper, or that may require crucial new experiments without which the paper should not be published. The originally submitted paper needed some work in clarifying specific and central conclusions that the authors draw, which the authors have done during revision.

---

## [Referee Report · Reviewer #2 (Public review)]

Summary:

An initial screening of pretreatment with different stress treatments of K. pneumonia allowed the identification of heat stress as a protection factor against the infection of the lytic phage Kp11. Then experiments prove that this is mediated not by an increase of phage resistant bacteria but due to an increase in phage transient tolerant population, that the authors identified as bacteriophage persistence in analogy to antibiotic persistence. Then they proved that phage persistence mediated by heath shock enhanced the evolution of bacterial resistance against the phage. The same trait was observed using other lytic phages, their combinations and two clinical strains, as well as *E. coli* and two T phages, hence the phenomenon may be widespread in enterobacteria.

Next, the elucidation of heat induced phage persistence was done, determining that phage adsorption was not affected but phage DNA internalization was impaired by the heat pretreatment, likely to alterations in the bacterial envelope, including the downregulation of envelope proteins and of LPS; furthermore, heat treated bacteria were less sensitive to polymyxins due to the decrease in LPS.

Finally, cyclic exposure to heat stress allowed the isolation of a mutant that was both resistant to heat treatment, polymyxins and lytic phage, that mutant had alterations in PspA protein that allowed a gain of function and that promoted the reduction of capsule production and loss of its structure; nevertheless this mutant was severely impaired in immune evasion as it was easily cleared from mice blood, evidencing the trade-off's between phage/heat and antibiotic resistance and the ability to counteract the immune response.

Strengths:

The experimental design and the sequence in which they are presented is ideal for the understanding of their study and the conclusions are supported by the findings, also the discussion points out the relevance of their work particularly in the effectiveness of phage therapy and allow the design of strategies to improve their effectiveness.

Weaknesses:

In its present form it lacks the incorporation of some relevant previous work that explored the role of heat stress in phage susceptibility, antibiotic susceptibility, trade offs between phage resistance and resistance against other kinds of stress, virulence, etc. and the fact that exposure to lytic phages induces antibiotic persistence.

Comments on revised version:

Thanks for addressing most of my comments; however, although you replied this in the rebuttal:

"Thank you for highlighting these important studies. We have incorporated the work by Majkowska-Skrobek et al. (2021), Gordillo Altamirano et al. (2021), and García-Cruz et al. (2024) into the discussion "

I was not able to find the new section in the discussion of the manuscript.

---

## [Author Response]

**Public Reviews:**

**Reviewer #1 (Public review):**
Summary:In this interesting and original paper, the authors examine the effect that heat stress can have on the ability of bacterial cells to evade infection by lytic bacteriophages. Briefly, the authors show that heat stress increases the tolerance of Klebsiella pneumoniae to infection by the lytic phage Kp11. They also argue that this increased tolerance facilitates the evolution of genetically encoded resistance to the phage. In addition, they show that heat can reduce the efficacy of phage therapy. Moreover, they define a likely mechanistic reason for both tolerance and genetically encoded resistance. Both lead to a reorganization of the bacterial cell envelope, which reduces the likelihood that phage can successfully inject their DNA.Strengths:I found large parts of this paper well-written and clearly presented. I also found many of the experiments simple yet compelling. For example, the experiments described in Figure 3 clearly show that prior heat exposure can affect the efficacy of phage therapy. In addition, the experiments shown in Figures 4 and 6 clearly demonstrate the likely mechanistic cause of this effect. The conceptual Figure 7 is clear and illustrates the main ideas well. I think this paper would work even without its central claim, namely that tolerance facilitates the evolution of resistance. The reason is that the effect of environmental stressors on stress tolerance has to my knowledge so far only been shown for drug tolerance, not for tolerance to an antagonistic species.Weaknesses:I did not detect any weaknesses that would require a major reorganization of the paper, or that may require crucial new experiments. However, the paper needs some work in clarifying specific and central conclusions that the authors draw. More specifically, it needs to improve the connection between what is shown in some figures, how these figures are described in the caption, and how they are discussed in the main text. This is especially glaring with respect to the central claim of the paper from the title, namely that tolerance facilitates the evolution of resistance. I am sympathetic to that claim, especially because this has been shown elsewhere, not for phage resistance but for antibiotic resistance. However, in the description of the results, this is perhaps the weakest aspect of the paper, so I'm a bit mystified as to why the authors focus on this claim. As I mentioned above, the paper could stand on its own even without this claim.

Thank you for your feedback. We understand your concern regarding the central claim that tolerance facilitates the evolution of resistance, while the paper can stand on its own without this claim, we think it provides an important layer to the interpretation of our findings. Considering your comments, we plan to revise the title and adjust to “Heat Stress Induces Phage Tolerance in Bacteria”.

More specific examples where clarification is needed:(1) A key figure of the paper seems to be Figure 2D, yet it was one of the most confusing figures. This results from a mismatch between the accompanying text starting on line 92 and the figure itself. The first thing that the reader notices in the figure itself is the huge discrepancy between the number of viable colonies in the absence of phage infection at the two-hour time point. Yet this observation is not even mentioned in the main text. The exclusive focus of the main text seems to be on the right-hand side of the figure, labeled "+Phage". It is from this right-hand panel that the authors seem to conclude that heat stress facilitates the evolution of resistance. I find this confusing, because there is no difference between the heat-treated and non-treated cells in survivorship, and it is not clear from this data that survivorship is caused by resistance, not by tolerance/persistence. (The difference between tolerance and resistance has only been shown in the independent experiments of Figure 1B.)

Thank you for your helpful comment. Figure 2d presents colony counts from a plating assay following the phage killing experiment in Figure 2c. Bacteria collected after 0 and 2 hours of phage exposure were plated on both phage-free (−phage) and phage-containing (+phage) plates. The “−phage” condition reflects total survivors, while the “+phage” condition indicates the resistant subset.

As seen in Figure 2d (left part), heat-treated bacteria showed markedly higher survival on phage-free plates than untreated cells, which were largely eliminated by phage. However, resistant colony counts on phage-containing plates were similar between two groups (as shown in figure 2d right part), suggesting that heat stress increased survival but did not promote resistance.

To clarify, we have revised the labels in Figure 2d as follows: “Total” will replace “-phage” to indicate the total survivors from the phage killing assay, and “Resisters” will replace “+phage” to indicate the resistant survivors, which are detected on phage-containing plates. This adjustment should eliminate any confusion and better reflect the experimental design.

Figure 2F supports the resistance claim, but it is not one of the strongest experiments of the paper, because the author simply only used "turbidity" as an indicator of resistance. In addition, the authors performed the experiments described therein at small population sizes to avoid the presence of resistance mutations. But how do we know that the turbidity they describe does not result from persisters?I see three possibilities to address these issues. First, perhaps this is all a matter of explaining and motivating this particular experiment better. Second, the central claim of the paper may require additional experiments. For example, is it possible to block heat induced tolerance through specific mutations, and show that phage resistance does not evolve as rapidly if tolerance is blocked? A third possibility is to tone down the claim of the paper and make it about heat tolerance rather than the evolution of heat resistance.

Thank you for your thoughtful comment. We appreciate the opportunity to clarify the interpretation of Figure 2f and the rationale behind the experimental design. We agree that turbidity alone cannot fully distinguish resistance from persistence. However, our earlier experiments (Figures 2d and 2e) demonstrated that heat-treated survivors remained largely susceptible to phage, indicating that heat stress does not directly induce resistance. This led us to hypothesize that heat enhances phage tolerance, which in turn increases the likelihood of resistance emergence during subsequent infection.

To test this, we used a low initial bacterial population (~10³ CFU per well) to minimize the chance of pre-existing resistance. Bacteria were exposed to phages at MOIs of 1, 10, and 100 and incubated for 24 hours in 100 µL volumes. This setup ensured:

(1) The low initial population minimizes the presence of pre-existing resistant mutants, ensuring that any phage-resistant bacteria observed arise during the infection process.

(2) The high MOI (≥ 1) ensures that each bacterial cell has a high probability of infection by at least one phage.

(3) The small volume (100 µL per well) maximizes the interaction between bacteria and phages, ensuring rapid infection of susceptible bacteria, which leads to clear wells. If resistant mutants arise, they will grow and cause turbidity.

Thus, the turbidity observed in heat-treated samples reflects de novo emergence and outgrowth of resistant mutants from a tolerant population. This assay supports the idea that heat-induced tolerance increases the probability of resistance evolution, rather than directly causing resistance.

We have revised the text to better explain this experimental logic and adjust the framing of our conclusions accordingly.

A minor but general point here is that in Figure 2D and in other figures, the labels "-phage" and "+phage" do not facilitate understanding, because they suggest that cells in the "-phage" treatment have not been exposed to phage at all, but that is not the case. They have survived previous phage treatment and are then replated on media lacking phage.

Thank you for your valuable comment. To clarify, we have revised the labels in Figure 2d as follows: “Total” will replace “-phage” to indicate the total survivors from the phage killing assay, and “Resisters” will replace “+phage” to indicate the resistant survivors, which are detected on phage-containing plates.

(2) Another figure with a mismatch between text and visual materials is Figure 5, specifically Figures 5B-F. The figure is about two different mutants, and it is not even mentioned in the text how these mutants were identified, for example in different or the same replicate populations. What is more, the two mutants are not discussed at all in the main text. That is, the text, starting on line 221 discusses these experiments as if there was only one mutant. This is especially striking as the two mutants behave very differently, as, for example, in Figure 5C. Implicitly, the text talks about the mutant ending in "...C2", and not the one ending in "...C1". To add to the confusion, the text states that the (C2) mutant shows a change in the pspA gene, but in Figure 5f, it is the other (undiscussed) mutant that has a mutation in this gene. Only pspA is discussed further, so what about the other mutants? More generally, it is hard to believe that these were the only mutants that occurred in the genome during experimental evolution. It would be useful to give the reader a 2-3 sentence summary of the genetic diversity that experimental evolution generated.

Thank you for your thoughtful comment. In our heat treatment evolutionary experiment, we isolated six distinct bacterial clones, of which two are highlighted in the manuscript as representative examples. One clone, BC2G11C1, acquired both heat tolerance and phage resistance, while another clone, BC3G11C2, became heat-tolerant but did not develop resistance to phage infection. This variation highlights the inherent diversity in evolutionary responses when exposed to selective pressures. It demonstrates that not all evolutionary pathways lead to the same outcome, even under similar stress conditions. This variability is a key observation in our study, illustrating that different genetic adaptations may arise depending on the specific mutations or genetic context, and not every strain will evolve phage resistance in parallel with heat tolerance. We have updated the manuscript to better reflect this diversity in the evolutionary trajectories observed.

**Reviewer #2 (Public review):**
Summary:An initial screening of pretreatment with different stress treatments of K. pneumoniae allowed the identification of heat stress as a protection factor against the infection of the lytic phage Kp11. Then experiments prove that this is mediated not by an increase of phage-resistant bacteria but due to an increase in phage transient tolerant population, which the authors identified as bacteriophage persistence in analogy to antibiotic persistence. Then they proved that phage persistence mediated by heat shock enhanced the evolution of bacterial resistance against the phage. The same trait was observed using other lytic phages, their combinations, and two clinical strains, as well as *E. coli* and two T phages, hence the phenomenon may be widespread in enterobacteria.Next, the elucidation of heat-induced phage persistence was done, determining that phage adsorption was not affected but phage DNA internalization was impaired by the heat pretreatment, likely due to alterations in the bacterial envelope, including the downregulation of envelope proteins and of LPS; furthermore, heat treated bacteria were less sensitive to polymyxins due to the decrease in LPS.Finally, cyclic exposure to heat stress allowed the isolation of a mutant that was both resistant to heat treatment, polymyxins, and lytic phage, that mutant had alterations in PspA protein that allowed a gain of function and that promoted the reduction of capsule production and loss of its structure; nevertheless this mutant was severely impaired in immune evasion as it was easily cleared from mice blood, evidencing the tradeoffs between phage/heat and antibiotic resistance and the ability to counteract the immune response.Strengths:The experimental design and the sequence in which they are presented are ideal for the understanding of their study and the conclusions are supported by the findings, also the discussion points out the relevance of their work particularly in the effectiveness of phage therapy and allows the design of strategies to improve their effectiveness.Weaknesses:In its present form, it lacks the incorporation of some relevant previous work that explored the role of heat stress in phage susceptibility, antibiotic susceptibility, tradeoffs between phage resistance and resistance against other kinds of stress, virulence, etc., and the fact that exposure to lytic phages induces antibiotic persistence.

Thank you for your insightful comments. I appreciate your suggestion regarding the inclusion of relevant previous works. I have now incorporated additional citations to discuss these points, including studies on the relationship between heat stress and antibiotic resistance, as well as the tradeoffs between phage resistance and other stress factors.

**Reviewer #3 (Public review):**
PspA, a key regulator in the phage shock protein system, functions as part of the envelope stress response system in bacteria, preventing membrane depolarization and ensuring the envelope stability. This protein has been associated in the Quorum Sensing network and biofilm formation. (Moscoso M., Garcia E., Lopez R. 2006. Biofilm formation by Streptococcus pneumoniae: role of choline, extracellular DNA, and capsular polysaccharide in microbial accretion. J. Bacteriol. 188:7785-7795; Vidal JE, Ludewick HP, Kunkel RM, Zähner D, Klugman KP. The LuxS-dependent quorum-sensing system regulates early biofilm formation by Streptococcus pneumoniae strain D39. Infect Immun. 2011 Oct;79(10):4050-60.)It is interesting and very well-developed.(1) Could the authors develop experiments about the relationship between Quorum Sensing and this protein?(2) It would be interesting to analyze the link to phage infection and heat stress in relation to Quorum. The authors could study QS regulators or AI2 molecules.

Thank you for your insightful comments and for bringing up the role of PspA in quorum sensing and biofilm formation. However, we would like to clarify a potential misunderstanding: the PspA discussed in our manuscript refers to phage-shock protein A, a key regulator in the bacterial envelope stress response system. This is distinct from the pneumococcal surface protein A, which has been associated with quorum sensing and biofilm formation in Streptococcus pneumoniae (as referenced in your comment).

To avoid any confusion for readers, we will ensure that our manuscript explicitly states “phage-shock protein A (PspA)” at its first mention. We appreciate your feedback and hope this clarification addresses your concern.

(3) Include the proteins or genes in a table or figure from lytic phage Kp11 (GenBank: ON148528.1).

Thank you for your helpful suggestion. We have now included a figure, as appropriate summarizing the proteins of the lytic phage Kp11 (GenBank: ON148528.1) in supplementary Figure S1.

**Recommendations for the authors:**

**Reviewer #1 (Recommendations for the authors):**
Issues unrelated to those discussed in the public review(1) Figure 4a and its caption describe an evolution experiment, but they do not mention how many cycles of high-temperature treatment and growth this experiment lasted. I assume it lasted for more than one cycle, because the methods section mentions "cycles", but the number is not provided.

Thank you for pointing this out. The evolutionary experiment shown in Figure 5a involved 11 cycles of high-temperature treatment and growth. We have now explicitly stated this in the figure legend to ensure clarity: BC: Batch culture, G: Evolution cycle number, C: Colony. BC2G11C1 refers to the first colony from batvh culture 2 after 11 rounds of heat treatment.

(2) It is not clear what Figure 5F is supposed to show. What are the gray boxes? The caption claims that the figure shows non-synonymous mutations, but the only information it contains is about genes that seem to be affected by mutation. Judging from the mismatch between the main text and the figure, the mutants with these mutations may actually be mislabeled.

Thank you for your careful review. Figure 5f highlights the non-synonymous mutations identified in the evolved strains. The gray boxes represent the ancestral strain’s whole genome without mutations, serving as a control. The corresponding labels indicate the specific mutations found in each evolved strain. We have clarified this in the figure caption to improve clarity. Additionally, we have carefully reviewed the labeling to ensure accuracy and consistency between the figure, main text, and sequencing data.

(3) I think that the acronym NC, which is used in just about every figure, is explained nowhere in the paper. Spell out all acronyms at first use.

Thank you for pointing this out. We have rivewed ensure that NC is clearly defined at its first mention in the text and figure legends to improve clarity. Additionally, we have reviewed the manuscript to ensure that all acronyms are properly introduced when first used.

(4) The same holds for the acronym N.D. This is an especially important oversight because N.D. could mean "not determined" or "not detectable", which would lead to very different interpretations of the same figure.

Thank you for your careful review. We have clarified the meaning of N.D., which stands for non-detectable, at its first use to avoid ambiguity and ensure accurate interpretation in the figure legend. Additionally, we have reviewed the manuscript to ensure that all acronyms are clearly defined.

(5) The panel labels (a,b, etc.) in all figure captions are very difficult to distinguish from the rest of the text, and should be better highlighted, for example by using a bold font. However, this is a matter of journal style and will probably be fixed during typesetting.

Thank you for your suggestion. We have adjusted the figure captions to better distinguish panel labels, such as using bold font, to improve readability and final formatting will follow the journal’s style during typesetting.

(6) Line 224: enhanced insusceptibility -> reduced susceptibility.

Thank you for your suggestion. We have revised “enhanced insusceptibility” to “reduced susceptibility” for clarity and precision.

(7) Line 259: mice -> mouse.

Thank you for catching this. We have corrected “mice” to “mouse”.

**Reviewer #2 (Recommendations for the authors):**
I have no concerns about the experimental design and conclusions of your work; however, I strongly recommend incorporating several relevant pieces of the literature related to your work, in the discussion of your manuscript, specifically:(1) Previous studies about the role of heat stress in phage infections, see:Greenrod STE, Cazares D, Johnson S, Hector TE, Stevens EJ, MacLean RC, King KC. Warming alters life-history traits and competition in a phage community. Appl Environ Microbiol. 2024 May 21;90(5):e0028624. doi: 10.1128/aem.00286-24. Epub 2024 Apr 16. PMID: 38624196; PMCID: PMC11107170.

Thank you for your thoughtful comment. We have ensured to incorporate the study by Greenrod et al. (2024) into the discussion to enrich the context of our findings. As this article pointed out, a temperature of 42°C can indeed limit phage infection in bacteria, acting as a barrier from the phage’s perspective. Our study builds on this by demonstrating that bacteria pre-treated with high temperatures exhibit tolerance to phage infection. These findings, together with the work you referenced, underscore the importance of heat stress or elevated temperature in host-phage interactions, with 42°C being particularly relevant in the context of fever. We will make sure to clarify this connection in our revised manuscript.

(2) The effect of heat stress and the tolerance/resistance against other antibiotics besides polymyxins, see:Lv B, Huang X, Lijia C, Ma Y, Bian M, Li Z, Duan J, Zhou F, Yang B, Qie X, Song Y, Wood TK, Fu X. Heat shock potentiates aminoglycosides against gram-negative bacteria by enhancing antibiotic uptake, protein aggregation, and ROS. Proc Natl Acad Sci U S A. 2023 Mar 21;120(12):e2217254120. doi: 10.1073/pnas.2217254120. Epub 2023 Mar 14. PMID: 36917671; PMCID: PMC10041086.

Thank you for bringing this study to our attention. We have incorporated the findings from Lv et al. (2023) into the discussion of our manuscript, highlighting how sublethal temperatures may facilitate the killing of bacteria by antibiotics like kanamycin. This is consistent with our data showing enhanced susceptibility of heat-shocked bacteria to kanamycin. The study also provides insights into the potential role of PMF, which is relevant to our work on PspA, and strengthens the broader context of heat stress influencing both antibiotic resistance and tolerance.

(3) Perhaps the most relevant overlooked fact was that recently it was demonstrated for *E. coli*, Klebsiella and Pseudomonas that pretreatment with lytic phages induced antibiotic persistence! Please discuss this finding and its implications for your work, see:Fernández-García L, Kirigo J, Huelgas-Méndez D, Benedik MJ, Tomás M, García-Contreras R, Wood TK. Phages produce persisters. Microb Biotechnol. 2024 Aug;17(8):e14543. doi: 10.1111/1751-7915.14543. PMID: 39096350; PMCID: PMC11297538.Sanchez-Torres V, Kirigo J, Wood TK. Implications of lytic phage infections inducing persistence. Curr Opin Microbiol. 2024 Jun;79:102482. doi: 10.1016/j.mib.2024.102482. Epub 2024 May 6. PMID: 38714140.

Thank you for suggesting this important reference. We agree that the phenomenon of phage-induced bacterial persistence is highly relevant to our study. While our manuscript focuses on the role of heat stress in bacterial tolerance and resistance, we acknowledge that bacterial persistence against phages is an established concept. We have incorporated this finding into our discussion, emphasizing how persistence and tolerance can overlap in their effects on bacterial survival, especially under stress conditions like heat treatment. This will provide a more comprehensive understanding of how phage interactions with bacteria can lead to both persistence and resistance.

(4) Finally, you observed a tradeoff pf the pspA* mutant increased phage/heat/polymyxin resistance and decreased immune evasion (perhaps by being unable to counteract phagocytosis), those tradeoffs between gaining phage resistance but losing resistance to the immune system, virulence impairment and resistance against some antibiotics had been extensively documented, see:Majkowska-Skrobek G, Markwitz P, Sosnowska E, Lood C, Lavigne R, Drulis-Kawa Z. The evolutionary trade-offs in phage-resistant Klebsiella pneumoniae entail cross-phage sensitization and loss of multidrug resistance. Environ Microbiol. 2021 Dec;23(12):7723-7740. doi: 10.1111/1462-2920.15476. Epub 2021 Mar 27. PMID: 33754440.Gordillo Altamirano F, Forsyth JH, Patwa R, Kostoulias X, Trim M, Subedi D, Archer SK, Morris FC, Oliveira C, Kielty L, Korneev D, O'Bryan MK, Lithgow TJ, Peleg AY, Barr JJ. Bacteriophage-resistant Acinetobacter baumannii are resensitized to antimicrobials. Nat Microbiol. 2021 Feb;6(2):157-161. doi: 10.1038/s41564-020-00830-7. Epub 2021 Jan 11. PMID: 33432151.García-Cruz JC, Rebollar-Juarez X, Limones-Martinez A, Santos-Lopez CS, Toya S, Maeda T, Ceapă CD, Blasco L, Tomás M, Díaz-Velásquez CE, Vaca-Paniagua F, Díaz-Guerrero M, Cazares D, Cazares A, Hernández-Durán M, López-Jácome LE, Franco-Cendejas R, Husain FM, Khan A, Arshad M, Morales-Espinosa R, Fernández-Presas AM, Cadet F, Wood TK, García-Contreras R. Resistance against two lytic phage variants attenuates virulence and antibiotic resistance in *Pseudomonas aeruginosa*. Front Cell Infect Microbiol. 2024 Jan 17;13:1280265. doi: 10.3389/fcimb.2023.1280265. Erratum in: Front Cell Infect Microbiol. 2024 Mar 06;14:1391783. doi: 10.3389/fcimb.2024.1391783. PMID: 38298921; PMCID: PMC10828002.

Thank you for highlighting these important studies. We have incorporated the work by Majkowska-Skrobek et al. (2021), Gordillo Altamirano et al. (2021), and García-Cruz et al. (2024) into the discussion to provide further context to the evolutionary trade-offs observed in our study. The findings in these studies, which describe the cross-sensitization to antimicrobials and the loss of multidrug resistance in phage-resistant bacteria, align with our observations of trade-offs in the *pspA* mutant. Specifically, our results show that while the *pspA* mutant exhibits increased resistance to phage, heat, and polymyxins, it also experiences a decrease in immune evasion and potential virulence. These trade-offs are significant in understanding the broader consequences of developing resistance to phages and other stressors.